# The Extracellular Matrix Vitalizer RA^TM^ Increased Skin Elasticity by Modulating Mitochondrial Function in Aged Animal Skin

**DOI:** 10.3390/antiox12030694

**Published:** 2023-03-11

**Authors:** Kyung-A Byun, Seyeon Oh, Sosorburam Batsukh, Min Jeong Kim, Je Hyuk Lee, Hyun Jun Park, Moon Suk Chung, Kuk Hui Son, Kyunghee Byun

**Affiliations:** 1Department of Anatomy & Cell Biology, Gachon University College of Medicine, Incheon 21936, Republic of Korea; 2Functional Cellular Networks Laboratory, Department of Medicine, Graduate School and Lee Gil Ya Cancer and Diabetes Institute, College of Medicine, Gachon University, Incheon 21999, Republic of Korea; 3Mihana Clinic, Gyeonggi 17051, Republic of Korea; 4Doctorbom Clinic, Seoul 06614, Republic of Korea; 5Maylin Anti-Aging Clinic, Seoul 06005, Republic of Korea; 6I’ll Global Co., Inc., Seoul 06532, Republic of Korea; 7Department of Thoracic and Cardiovascular Surgery, Gachon University Gil Medical Center, Gachon University, Incheon 21565, Republic of Korea

**Keywords:** basement membrane, extracellular matrix vitalizer-RA^TM^, mitochondrial dysfunction, oxidative stress, skin rejuvenation

## Abstract

Oxidative stress-induced cellular senescence and mitochondrial dysfunction result in skin aging by increasing ECM levels-degrading proteins such as MMPs, and decreasing collagen synthesis. MMPs also destroy the basement membrane, which is involved in skin elasticity. The extracellular matrix vitalizer RA^TM^ (RA) contains various antioxidants and sodium hyaluronate, which lead to skin rejuvenation. We evaluated whether RA decreases oxidative stress and mitochondrial dysfunction, eventually increasing skin elasticity in aged animals. Oxidative stress was assessed by assaying NADPH oxidase activity, which is involved in ROS generation, and the expression of SOD, which removes ROS. NADPH oxidase activity was increased in aged skin and decreased by RA injection. SOD expression was decreased in aged skin and increased by RA injection. Damage to mitochondrial DNA and mitochondrial fusion markers was increased in aged skin and decreased by RA. The levels of mitochondrial biogenesis markers and fission markers were decreased in aged skin and increased by RA. The levels of NF-κB/AP-1 and MMP1/2/3/9 were increased in aged skin and decreased by RA. The levels of TGF-β, CTGF, and collagen I/III were decreased in aged skin and increased by RA. The expression of laminin and nidogen and basement membrane density were decreased in aged skin and increased by RA. RA increased collagen fiber accumulation and elasticity in aged skin. In conclusion, RA improves skin rejuvenation by decreasing oxidative stress and mitochondrial dysfunction in aged skin.

## 1. Introduction

The primary triggering factor of skin aging is oxidative cellular damage caused by increased oxidative stress [1,2]. Oxidative stress results from an imbalance between reactive oxygen species (ROS) synthesis and defense mechanisms that remove ROS [1,2]. Enzymes that remove ROS, such as glutathione (GSH), superoxide dismutase (SOD), and catalase, are representative of defense mechanisms against oxidative stress [3].

During the skin aging process, oxidative stress leads to the upregulation of nuclear factor kappa-light-chain-enhancer of activated B cells (NF-κB) and its downstream signal pathway of activator protein 1 (AP-1), which eventually increases the levels of extracellular matrix (ECM)-degrading proteins such as collagenase and matrix metalloproteinases (MMPs) [4,5,6].

ROS also decreases collagen synthesis by downregulating the transforming growth factor-β (TGF-β) pathway [7]. In addition, ROS downregulates type II TGF-β receptor and its downstream signaling pathways of mothers against decapentaplegic homolog 3 and connective tissue growth factor (CTGF), eventually leading to decreased synthesis of type I collagen [7].

Increased ROS levels damage mitochondrial DNA (mtDNA), which leads to mitochondrial dysfunction [8,9,10]. Increased oxidative stress induced by H_2_O_2_ treatment in dermal fibroblasts leads to decreased levels of peroxisome proliferator-activated receptor γ coactivator 1 α (PGC-1α), which is a significant modulator of mitochondrial biogenesis [11]. Moreover, the PGC-1α level decreases with aging in various tissues [12]. Damage to mtDNA in human keratinocytes caused by UV radiation was accompanied by increased levels of collagen degradation enzymes and ROS synthesis, which aggravated mitochondrial dysfunction [13,14,15].

Mitochondrial dynamics alter mitochondrial morphology via fusion and fission and are essential for maintaining mitochondrial number and shape [16]. Mitofusin (MFN) 1 and 2 and optic atrophy protein 1 (OPA1) are required for mitochondrial fusion, which provides a trade-off between damaged mtDNA and healthy mtDNA [16]. Mitochondrial fission, performed by dynamin-related protein 1 (DRP1) or mitochondrial fission 1 protein (FIS1), leads to the formation of new mitochondria [16].

During cellular senescence, mitochondrial fission is decreased, which leads to mitochondrial elongation [17,18]. Increased oxidative stress decreases the levels of fission proteins such as FIS1 [17]. Oxidative stress induced by UV radiation resulted in increased levels of fusion proteins such as OPA1 and MFN2 and decreased levels of DRP1 [19]. Mitochondrial dysfunction leads to dermal fibroblast and keratinocyte senescence, which causes skin wrinkle formation via ECM changes [20,21,22,23,24,25].

The epidermal basement membrane (BM) in the dermal-epidermal junction has a sheet-like structure and acts as a binder between the dermis and the epidermis [26]. The major proteins of the BM are type IV and type VII collagens, nidogen, laminins, and perlecan [26]. Since the MMPs destroy the BM, radiation causes BM degradation [27,28]. Moreover, the BM is thinned, accompanied by decreased BM protein gene expression during aging [29,30]. The expression of a structural protein, collagen IV, is reduced in aged skin or senescent fibroblasts [31]. In contrast, laminin or nidogen-stimulating peptide complexes increased dermal collagen XVII and laminin levels in excised human skin and decreased wrinkles in Asian females [32].

Various effective compounds, such as vitamins, amino acids, hyaluronic acid, and minerals, have been used to rejuvenate aged skin. Since ascorbic acid (AA, vitamin C), a well-known antioxidant, is an essential factor for the synthesis of DNA and collagen, it has been frequently used for skin rejuvenation [33,34]. Niacinamide (NA, vitamin B3) is a precursor of nicotinamide adenine dinucleotide [35], which is a redox cofactor [12,36]. Since NA shows antioxidative and anti-inflammatory effects, it has also been used for skin rejuvenation [37]. Glutathione is a powerful antioxidant and decreases melasma when used as a mesotherapy [38]. Since hyaluronic acid can hold 1000 times its own weight of water, it is beneficial for skin hydration [39]. Moreover, hyaluronic acid showed anti-inflammatory and antioxidant effects [40,41]. In aged skin, dermal fibroblasts show decreased synthesis of hyaluronic acid [42]. Hyaluronic acid complex with various vitamins increased dermal fibroblast proliferation [42].

The extracellular matrix vitalizer RA^TM^ (RA, illglobal, Seoul, Republic of Korea) contains various antioxidants, such as AA, NA, coenzymes, glutathione, and sodium hyaluronate. Thus, we hypothesized that RA injection could decrease oxidative stress and cellular senescence, which eventually resulted in a reduction in NF-κB/AP-1 and mitochondrial dysfunction in the skin. These reductions led to decreased levels of MMPs, which eventually decreased ECM fiber destruction and BM destruction. RA also increased the levels of TGF-β and CTGF, which eventually increased collagen fiber synthesis. We evaluated the RA-mediated increase in collagen fiber accumulation and decrease BM destruction via decreased oxidative stress in aged animals. The effects of RA were compared with those of AA or NA injected alone into aged animal’s skin.

## 2. Materials and Methods

### 2.1. Preparation of RA

RA was formulated as a liquid before application. First, AA, NA, coenzymes, glutathione, and sodium hyaluronate were dissolved in distilled water with mixing at 3000 rpm using a high-speed mixer (T.K. Homo Disper, Model 2.5, PRIMIX, Hyogo, Japan). Then, the RA solution was filtered through a 0.2 μm filter (S2GPU11RE, Merck, Darmstadt, Land Hessen, Germany) to remove bacteria. The RA liquid contained 0.25% AA and 0.25% NA (Appendix A).

### 2.2. In Vitro Model

#### 2.2.1. Cell Culture

Human primary epidermal keratinocytes (HEKn; ATCC, Manassas, VA, USA) were cultivated with dermal cell basal medium (ATCC) with a keratinocyte growth kit (ATCC) and maintained at 37 °C under 5% CO_2_.

#### 2.2.2. NA, AA, and RA Treatment

To determine whether keratinocytes were affected by NA, AA, or RA treatment, HEKn cells were treated with 50 μM H_2_O_2_ for 2 h, treated with NA (0.4 mM), AA (0.4 mM), or RA (80 μL), and cultured for 48 h (Appendix A). In addition, control cells were treated with phosphate-buffered saline (PBS).

### 2.3. In Vivo Model

#### 2.3.1. Mouse Conditions

Eight-week-old male C57BL/6 mice were obtained from Orient Bio (Seongnam, Korea). The young group contained 9-week-old mice after one week of acclimatization, and the aging group was bred until 12 months old.

This study was approved by the ethical board of the Center for Animal Care and Use. It was conducted by the guidelines of the Institutional Animal Care and Use Committee of Gachon University (approval number: LCDI-2022-0095). The mice used in this study were domesticated in an area with controlled temperature (22 ± 5 °C), relative humidity (50 ± 10%), and a 12-h light-dark cycle. In addition, they had free access to standard laboratory diets and water.

#### 2.3.2. RA Treatment

To determine whether aged animal skin was affected by RA treatment, 12-month-old aging mice were injected intradermally with RA (100 μL/cm^2^/day) twice every two weeks using a microneedle therapy system (MTS; Derma-Q Gold 0.5 mm, DONGBANG medicare, Seongnam, Korea). The control was injected with distilled water under the same conditions (Appendix A).

#### 2.3.3. Skin Elasticity

To confirm whether the skin elasticity of aged animal skin was changed by RA treatment, the skin elasticity of the animal before RA treatment and after 4 weeks was measured. Skin elasticity was evaluated with API-100^®^ (Aram Huvis Co., Ltd., Seongnam, Republic of Korea), and the average was used after measuring 5 times for each animal.

### 2.4. Sample Preparation

#### 2.4.1. Protein Isolation

Proteins were isolated from the cells and skin tissues by using the EzRIPA lysis kit (ATTO Corporation, Tokyo, Japan). First, the cells and skin tissues were lysed with EzRIPA buffer containing protease and phosphatase inhibitors. Then, the lysed samples were sonicated and centrifuged at 14,000× *g* for 20 min at 4 °C. Then, the supernatants were transferred to a new tube, and the protein was quantified by using a bicinchoninic acid assay kit (Thermo Fisher Scientific, Waltham, MA, USA).

#### 2.4.2. RNA Extraction and cDNA Synthesis

The total RNA from cells and frozen skin tissues was extracted using RNAiso Plus (Takara Bio, Kusatsu, Japan) according to the manufacturer’s instructions. The quality and concentration of the extracted RNA were confirmed by a Nanodrop 2000 spectrophotometer (Thermo Fisher Scientific), and cDNA was synthesized by using a PrimeScript First Strand cDNA Synthesis Kit (Takara Bio) according to the manufacturer’s instructions.

#### 2.4.3. Paraffin-Embedded Tissue

The harvested skin tissues were fixed with cold 4% paraformaldehyde (Sigma-Aldrich, St. Louis, MO, USA) in PBS at 4 °C for 24 h. The fixed skin tissues were washed for 30 min, and a paraffin block was made using a tissue processor (Thermo Fisher Scientific). The paraffin blocks were sectioned to 7 µm in thickness using a microtome (Leica Biosystems, Nussloch, Germany) and dried at 60 °C overnight to allow them to attach to the slides.

### 2.5. Nicotinamide Adenine Dinucleotide Phosphate (NADPH) Oxidase and SOD Activity

NADPH oxidase (Abcam, Cambridge, UK) and SOD (Abcam) activities in the H_2_O_2_-treated HEKn cells and skin tissue of each group were determined by using appropriate kits, following the manufacturers’ instructions.

### 2.6. Enzyme-Linked Immunosorbent Assay (ELISA)

To measure the levels of the 8-hydroxy-2′-deoxyguanosine (8-OHdG), Collagen type I alpha 1 (COL1A1), and Collagen type III alpha 1 (COL3A1), 96-well microplates were coated in 100 nM carbonate and bicarbonate-mixed buffer, adjusted to pH 9.6 and incubated overnight at 4 °C. The microplates were then washed with PBS containing 0.1% Tween 20 (TPBS). The remaining protein-binding sites were blocked using 5% skim milk for 6 h at room temperature. After washing with PBS, 30 μg of protein samples were distributed into each well and incubated overnight at 4 °C. Each well was rinsed with TPBS and then incubated with primary antibodies diluted in PBS overnight at 4 °C (Appendix A). After washing, peroxidase-conjugated secondary antibodies (Vector Laboratories, Newark, CA, USA) was loaded for 4 h at room temperature. Tetramethylbenzidine solution was added, followed by incubation for 15–20 min at room temperature. The stop solution that was used was 2 N sulfuric acid. The optical density was measured at a wavelength of 450 nm using a microplate reader (Molecular Devices, San Jose, CA, USA).

### 2.7. Western Blotting

Equal amounts of proteins were separated on 8–12% polyacrylamide gels and transferred to polyvinylidene fluoride membranes (Millipore, Burlington, Massachusetts, USA) by a power station (ATTO). After blocking using 5% skim milk and washing with Tris-buffered saline containing 0.1% Tween 20 (TTBS), we incubated the membranes with primary antibodies (Appendix A) for 15 h at 4 °C and then washed them with TTBS. Next, the membranes were incubated with peroxidase-conjugated secondary antibodies (Vector Laboratories) at room temperature for 1 h and rinsed with TTBS. Subsequently, an enhanced chemiluminescence detection reagent (Cytiva^TM^, Seoul, Korea) and imaging system (ChemiDoc; Bio-rad, Hercules, CA, USA) were used to visualize the immunoreactive proteins on the membrane.

### 2.8. Quantitative Real-Time Polymerase Chain Reaction (qRT–PCR)

The qRT–PCR reagent mixture was prepared by mixing 1 µg of synthesized cDNA, SYBR Green reagent (Takara), and 10 pmol of primer (Appendix A). This mixture was added to a 384-well multi-plate and analyzed with a CFX386 Touch Real-Time PCR System (Bio-Rad).

### 2.9. Immunohistochemistry

The sectioned slides were passed through a series of xylene and ethanol solutions (100%, 90%, 80%, 70%) to remove the paraffin and then hydrated with distilled water and PBS. In order to reduce nonspecific binding, the slides were incubated with normal serum. The blocked slides were incubated with primary antibody (Appendix A) for 15 h at 4 °C and 1 h at room temperature. The slides were then rinsed with PBS and incubated with biotinylated secondary antibodies (Vector Laboratories) for 1 h at room temperature. The slides were again rinsed with PBS and then incubated with ABC reagent (Vector Laboratories) according to the manufacturer’s instructions. After washing with PBS, the slides were developed for 5 min using a 3,3′-diaminobenzidine tetrahydrochloride hydrate (DAB; Sigma-Aldrich). Then, the slides were washed with PBS, followed by distilled water, and counterstained with hematoxylin solution (DAKO, Glostrup Kommune, Denmark). After the slides were washed, they were dehydrated with absolute alcohol and mounted using xylene and dibutyl phthalate in xylene (DPX; Sigma-Aldrich). Images of the stained slides were acquired with an optical microscope (Olympus Optical Co., Tokyo, Japan), and the intensity was analyzed using ImageJ software (NIH, Bethesda, MD, USA).

### 2.10. Histological Analysis

#### 2.10.1. Periodic Acid-Schiff (PAS) Staining

After deparaffinization, skin tissues were incubated in 0.5% periodic acid (BBC Biochemical, McKinney, TX, USA) for 5 min and rinsed with distilled water. Then, they were incubated in Schiff’s reagent for 15 min and rinsed with running tap water. After they were incubated with hematoxylin for 2 min, the samples were rinsed with distilled water, dehydrated, mounted with DPX mount solution (Sigma-Aldrich), and observed under an optical microscope (Olympus Optical Co.) equipped with a slide scanner (Motic, Vancouver, British Columbia, Canada). All images were analyzed for collagen fiber density using ImageJ software (NIH).

#### 2.10.2. Masson’s Trichrome Staining

After deparaffinization, skin tissues were incubated in Bouin solution (Scytek Laboratories, West Logan, UT, USA) at 60 °C for 1 h and rinsed with distilled water. The sections were then placed in a working solution of iron hematoxylin (Scytek Laboratories) for 5 min, Biebrich scarlet acid fuchsin solution (Scytek Laboratories) for 5 min, phosphomolybdic-phosphotungstic acid solution (Scytek Laboratories) for 12 min, and aniline blue solution (Scytek Laboratories) for 3 min. The stained slides were mounted with DPX mount solution (Sigma-Aldrich) and observed under an optical microscope (Olympus Optical Co.) equipped with a slide scanner (Motic). All images were analyzed for collagen fiber density using ImageJ software (NIH).

### 2.11. Transmission Electron Microscopy (TEM)

Specimens were fixed for 12 h in 2% glutaraldehyde/2% paraformaldehyde in 0.1 M phosphate buffer (pH 7.4) and washed in 0.1 M phosphate buffer, postfixed with 1% OsO_4_ in 0.1 M phosphate buffer for 2 h, dehydrated with an ascending ethanol series (50%, 60%, 70%, 80%, 90%, 95%, 100%, and 100%) for 10 min each, and infiltrated with propylene oxide for 10 min.

The fixed samples were embedded using a Poly/Bed 812 kit (Polysciences, Warrington, PA, USA) and polymerized in an electron microscope oven (DOSAKA, Katsumi, Japan) at 65 °C for 12 h. The block was equipped with a diamond knife in an ultramicrotome, cut into 200 nm sections, and stained with toluidine blue for optical microscopy.

The region of interest was then cut into 80 nm sections using the ultramicrotome, placed on copper grids, double stained with 3% uranyl acetate for 30 min and 3% lead citrate for 7 min, and observed by TEM (JEOL, Tokyo, Japan) equipped with a Megaview III CCD camera (Olympus Optical Co.) at an acceleration voltage of 80 kV.

### 2.12. Statistical Analysis

All results are presented as the means ± standard deviations, and all statistical analyses were performed using SPSS version 22 (IBM Corporation; Armonk, NY, USA). Statistical significance was determined by the Kruskal–Wallis test for comparisons of each group, followed by a post hoc Mann–Whitney U test. In this study, groups marked with different letters indicate significant intergroup differences.

*, PBS/PBS-treated keratinocyte vs. H_2_O_2_/PBS-treated keratinocyte or Aging/MTS vs. Young/MTS

$, H_2_O_2_-treated keratinocytes vs. H_2_O_2_/PBS-treated keratinocyte or Aging/RA+MTS vs. Aging/MTS

#, vs. H_2_O_2_/RA-treated keratinocyte

## 3. Results

### 3.1. RA Decreased Oxidative Stress in H_2_O_2_-Treated Keratinocytes and Aged Animal Skin

First, we evaluated whether RA decreased oxidative stress in the cellular senescence model and aged animal skin. Since the H_2_O_2_-induced cellular senescence model is the most widely used in vitro aging model [43], we treated human keratinocytes with H_2_O_2_ (Appendix A).

After treating keratinocytes with H_2_O_2_, PBS, AA, NA, and RA were administered (Appendix A). The oxidative stress alleviation effect of RA was compared with that of PBS, AA, and NA. NADPH oxidases are one of the primary ROS-generating enzymes [44].

NADPH oxidase activity was increased by treatment with H_2_O_2_. It was decreased by the administration of AA, NA, and RA. The most prominent decrease was observed in the RA-treated keratinocytes (Figure 1A). SOD activity was decreased by treatment with H_2_O_2_. It was increased by the administration of AA, NA, and RA. The most prominent increase was observed in the RA-treated keratinocytes (Figure 1B).

8-OHdG is a widely used marker of oxidative damage to DNA [45]. The expression of 8-OHdG was increased by treatment with H_2_O_2._ It was decreased by the administration of AA, NA, and RA. The most prominent decrease was observed in the RA-treated keratinocytes (Figure 1C).

mtDNA damage was significantly increased by H_2_O_2_ treatment. It was decreased by the administration of AA, NA, and RA. The most prominent decrease was observed in the RA-treated keratinocytes (Figure 1D).

RA was injected two times into aged animal skin with an MTS every two weeks (Appendix A). The NADPH oxidase activity in aged animal skin was higher than that in young animal skin. It was decreased by RA injection (Figure 1E).

The SOD level, which was evaluated with western blotting, decreased in aged skin compared with young skin. It was increased by RA injection (Figure 1F,G).

The level of mtDNA damage in aged skin was higher than that in young skin, and it was decreased by RA injection. The 8-OHdG level in aged skin was higher than that in young skin, and it was decreased by RA injection (Figure 1H,I).

The findings indicated that RA decreased oxidative stress and oxidative stress-induced DNA damage.

### 3.2. RA Decreased Mitochondrial Dysfunction and Cellular Senescence in H_2_O_2_-Treated Keratinocytes and Aged Skin

PGC-1α, which is an essential controller of mitochondrial biogenesis [11], was decreased by treatment with H_2_O_2_. It was increased by the administration of AA, NA, and RA. The most prominent increase was observed in the RA-treated keratinocytes (Figure 2A).

Cytochrome c oxidase (COX)1 and succinate dehydrogenase complex, and flavoprotein subunit A (SDHA) are also mitochondrial biogenesis markers [46]. COX1 and SDHA levels were decreased by treatment with H_2_O_2_ and increased by administration of AA, NA, and RA. The most prominent increase was observed in the RA-treated keratinocytes (Figure 2B,C).

The levels of the mitochondrial fission markers DRP1 and FIS1 were decreased by treatment with H_2_O_2_ and increased by administration of AA, NA, and RA. The most prominent increase was observed in the RA-treated keratinocytes (Figure 2D,E).

The levels of the mitochondrial fusion markers OPA1 and MFN2 were increased by treatment with H_2_O_2_ and decreased by administration of AA, NA, and RA. The most prominent decrease was observed in the RA-treated keratinocytes (Figure 2F,G).

The levels of the cellular senescence markers P21 and P16 [47] were increased by treatment with H_2_O_2_ and decreased by administration of AA, NA, and RA. The most prominent decrease was observed in the RA-treated keratinocytes (Figure 2H,I).

The levels of the markers of mitochondrial biogenesis, PGC-1α, COX1, and SDHA, were decreased in aged skin and were increased by RA injection (Figure 3A–D).

The protein expression levels of DRP1 and FIS1 were decreased in aged skin and increased by RA injection (Figure 3E–G).

The protein expression levels of OPA1 and MFN2 were increased in aged skin and decreased by RA injection (Figure 3E,H,I).

The expression of P21 and P16 was increased in aged skin and decreased by RA injection (Figure 3J,K).

### 3.3. RA Decreased NF-Κb/AP-1 and MMP1/2/3/9 Expression in Aged Skin

The expression of NF-κB and AP-1 in aged skin was increased compared with that in young skin. However, these levels were decreased by RA injection (Figure 4A,B).

The expression of MMP1/2/3/9 in aged skin was increased compared with that in young skin. These levels were decreased by RA injection (Figure 4C–F).

### 3.4. RA Increased the Expression of Laminin and Nidogen and the BM Density

The expression of laminin and nidogen was significantly lower in aged skin than in young skin. These levels were increased by RA injection (Figure 5A–C).

The BM density was evaluated with PAS staining. The intensity of the pink color observed by PAS staining of the aged skin was lower than that of young skin; however, this intensity was increased by RA injection (Figure 5D,E).

The BM consists of three layers: the lamina lucida, lamina densa, and lamina fibroreticularis [48]. The lamina densa has a sheet-like structure. The lamina lucida exists between the lamina densa and the epithelial layer and forms hemidesmosomes, which are electron-dense plaques [27]. It is known that photoaging induced disruption and duplication of the lamina densa [29].

The hemidesmosomes and lamina densa were observed by transmission electron microscopy. In aged skin, the lamina densa was more disrupted, and the number of hemidesmosomes was less than that in young skin. By RA injection, disruption of the lamina densa was improved, and the number of hemidesmosomes was increased (Figure 5F).

### 3.5. RA Upregulated the Expression of TGF-Β, CTGF, and α-Smooth Muscle Actin (A-SMA) and Collagen Fiber Accumulation in Aged Skin

In aged skin, the expression of TGF-β and CTGF was decreased compared with that in young skin. These levels were increased by RA injection (Figure 6A–C).

When fibroblasts are changed to activated myofibroblasts that express α-SMA, the expression of collagen type I and collagen type III increases [49,50].

The expressions of α-SMA, COL1A1, and COL3A1 were decreased in aged skin compared with young skin and increased by RA (Figure 6A,D,E,F).

The collagen fiber density in the skin was evaluated with Masson’s trichrome staining. The collagen fiber density in aged skin was decreased compared with that in young skin, and this density was increased by RA injection (Figure 6G,H).

Skin elasticity was evaluated with API-100^®^ (Aram Huvis Co., Ltd.). Skin elasticity change was decreased in aged skin and was increased by RA injection (Figure 6I).

## 4. Discussion

During intrinsic (chronological) aging, increasing levels of senescent epidermal or dermal cells lead to aggravation of DNA damage and mitochondrial dysfunction in neighboring cells [51]. Moreover, environmental factors such as UV radiation lead to increased deterioration of cellular senescence [52,53].

Collagen fibers play an essential role in maintaining skin elasticity by supporting the skin matrix [54,55]. Type I collagen is the main type of skin collagen and is mainly produced by fibroblasts in the dermis layer [56]. During aging, the ability of fibroblasts to synthesize collagen decreases by 1.0–1.5% each year, and decreasing collagen is accompanied by the formation of wrinkles [57,58].

During skin aging, changes in communication between keratinocytes and fibroblasts result in decreased levels of collagen fibers in the skin by both decreasing collagen synthesis and increasing collagen destruction [59,60,61]. Aged keratinocytes lead to more destruction of elastin fiber than young keratinocytes after UV exposure [62].

For skin rejuvenation, various effective formulas, including antioxidants, have been used via various delivery systems, such as topical creams, hypodermic needles, and microneedles [63]. Since topical creams only spread following the skin surface, the penetrating amount of drug is just 10–20% of the total amount of drug included in the cream [64]. Hypodermic needles can deliver almost 90–100% of the contained drug; however, they cause pain [65,66]. The delivery efficacy of microneedles is similar to that of hypodermic needles; however, microneedles do not cause pain [66]. Microneedles penetrate the stratum corneum and deliver drugs to the epidermis or upper dermal layer [66].

Since we delivered RA via MTS, we thought that RA could first affect keratinocytes in the epidermis and then keratinocyte-modulated fibroblasts in the dermis. Thus, we evaluated whether RA could decrease oxidative stress and mitochondrial dysfunction in H_2_O_2_-induced senescent keratinocytes.

We evaluated oxidative stress by measuring NADPH oxidase activity and SOD activity in H_2_O_2_-treated keratinocytes. After the H_2_O_2_ treatment, NADPH oxidase activity was increased, and SOD activity was decreased. RA decreased NADPH oxidase activity and increased SOD activity. mtDNA damage and 8-OHdG levels, which are markers of oxidative damage to DNA, were increased by H_2_O_2_ treatment. However, they were decreased by RA.

Similar to the result of the in vitro test, the NADPH oxidase activity of aged skin was increased compared with that of young skin. SOD expression was decreased in aged skin compared with young skin. After RA injection, NADPH oxidase activity decreased, and SOD expression was increased.

Endogenous ROS are mainly generated in the mitochondria since ROS are a byproduct of energy production [67,68]. During aging, chronic accumulation of ROS in the mitochondria leads to mutations in mtDNA that cause mitochondrial dysfunction [69,70,71].

Since mitochondrial dysfunction causes senescence in both keratinocytes and fibroblasts, which eventually leads to skin wrinkle formation [20,21,22,23,24,25], we evaluated whether RA could modulate ECM destruction conditions by decreasing keratinocyte senescence and mitochondrial dysfunction.

After H_2_O_2_ treatment, the levels of mitochondrial biogenesis markers (PGC-1α, COX1, and SDHA) were decreased, and these levels were increased by RA administration. Mitochondrial fission, which was evaluated by measuring the expression of DRP1 and FIS1, was decreased by H_2_O_2_ treatment. The levels of these markers were increased by RA. Mitochondrial fusion, which was evaluated by measuring the expression of OPA1 and MFN2, was increased by H_2_O_2_ treatment and decreased by RA. The levels of the cellular senescence markers P21 and P16 were increased by H_2_O_2_ and decreased by RA. It seemed that H_2_O_2_ induced mitochondrial dysfunction and cellular senescence in keratinocytes, and this effect was reduced by RA.

These changes were also observed in aged animal skin. Mitochondrial biogenesis and fission were decreased in aged skin and increased by RA. In contrast, mitochondrial fusion was increased in aged skin and decreased by RA. Cellular senescence in aged skin, which was evaluated by measuring P21 and P16, also increased and was decreased by RA injection.

NF-κB/AP-1/MMPs, which are involved in one of the main pathways involved in ECM destruction during aging [4,5,6], were evaluated in aged skin. The expression of NF-κB and AP-1 was increased in aged skin and decreased by RA injection. MMP1/2/3/9 levels were increased in aged skin and decreased by RA injection.

MMPs destroy the BM as well as collagen fibers in the ECM of the dermis [72]. During aging, the levels of proteins that form the BM are decreased [73]. Proteins that form the structure of the BM, such as collagen XVII, are also involved in transmembrane signal transduction during keratinocyte differentiation [74]. Laminin is also involved in supporting structural stability as well as modulation of cellular proliferation, migration, and differentiation [74]. Since BM plays an essential role in skin homeostasis, controlling the levels of BM proteins that are decreased by aging has been considered a method for decreasing skin wrinkles [75].

Epidermal keratinocytes express laminin and nidogen and secrete those proteins into the dermal-epidermal junction [76]. Then, laminin and nidogen are assembled into the BM [76]. Thus, keratinocyte function for modulating the BM is also important, in addition to fibroblast function.

We hypothesized that senescent keratinocytes might affect the expression of BM proteins such as laminin and nidogen by increasing oxidative stress. Thus, we evaluated the expression of nidogen and laminin in aged skin. These levels were decreased in the aged skin and increased by RA injection. The BM density, which was evaluated with PAS staining, was also decreased in aged skin and increased by RA injection.

Since increased oxidative stress also leads to decreased collagen synthesis by decreasing TGF-β [77], we evaluated the levels of TGF-β and CTGF, which are involved in collagen synthesis. In aged skin, the levels of TGF-β and CTGF were decreased, and these levels were increased by RA injection. Moreover, the expression of COLI and COLIII and collagen density were decreased in aged skin and increased by RA injection. Skin elasticity, which was evaluated with API-100, was decreased in aged skin and increased by RA.

## 5. Conclusions

Our study showed that RA decreased oxidative stress and mtDNA injury, which eventually decreased mitochondrial dysfunction in aged skin. RA also decreased ECM destruction by decreasing the levels of NF-κB/AP-1/MMPs and increasing the levels of BM proteins such as nidogen and laminin. RA enhanced the collagen synthesis-related signaling pathway of TGF-β and CTGF. These modulations associated with RA treatment led to increased collagen fiber accumulation and skin elasticity in aged skin.

## Figures and Tables

**Figure 1 antioxidants-12-00694-f001:**
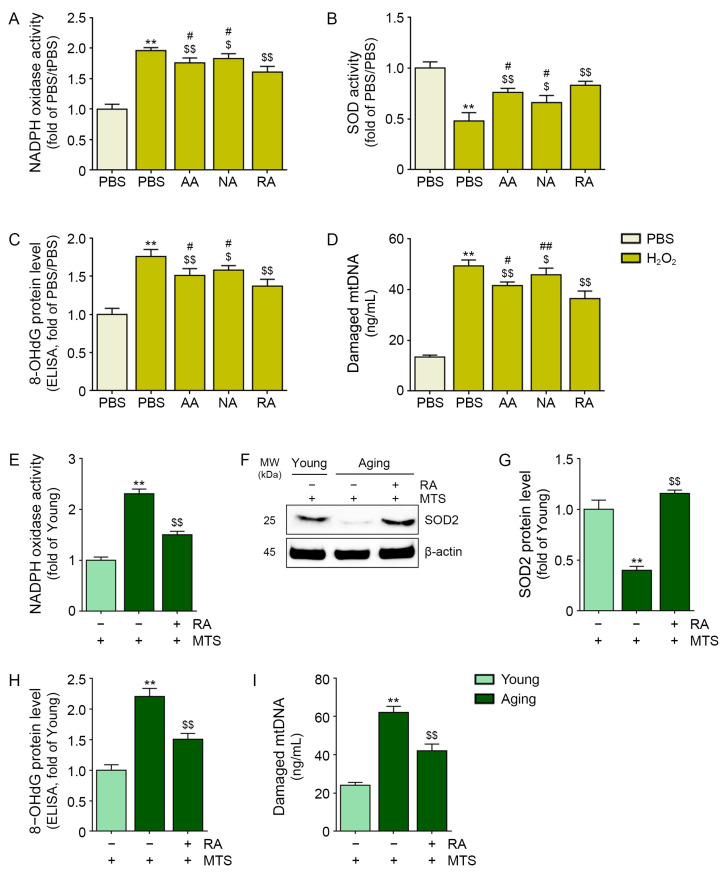
RA treatment decreased oxidative stress in H_2_O_2_-treated keratinocytes and aged mouse skin. (**A**,**E**) The NADPH oxidase activity in H_2_O_2_-treated keratinocytes (**A**) and aged mouse skin (**E**) was detected by an assay. The NADPH oxidase activity increased in H_2_O_2_-treated keratinocytes and aged mouse skin but decreased after RA treatment. (**B**,**F**,**G**) The SOD level in H_2_O_2_-treated keratinocytes (**B**) and aged mouse skin (**F**,**G**), as confirmed by the activity assay and western blot. The SOD level decreased in H_2_O_2_-treated keratinocytes and aged mouse skin but increased after RA treatment. (**C**,**H**) The 8-OHdG level in H_2_O_2_-treated keratinocytes (**C**) and aged mouse skin (**H**) was confirmed by ELISA. (**D**,**I**) The presence of damaged mtDNA in H_2_O_2_-treated keratinocytes (**D**) and aged mouse skin (**I**) was confirmed by an assay. The 8-OHdG and damaged mtDNA levels were increased in H_2_O_2_-treated keratinocytes and aged mouse skin but decreased by RA treatment. The data are presented as the mean ± SD of three independent experiments. **, *p* < 0.01, second bar vs. first bar; $, *p* < 0.05 and $$, *p* < 0.01, vs. second bar; #, *p* < 0.05 and ##, *p* < 0.01, vs. fifth bar (Mann–Whitney U test). 8-OHdG; 8-hydroxy-2′-deoxyguanosine, AA; ascorbic acid, ELISA; enzyme-linked immunosorbent assay, mtDNA; mitochondrial DNA, MW; molecular weight, NA; niacinamide, NADPH; nicotinamide adenine dinucleotide phosphate, PBS; phosphate-buffered saline, SOD; superoxide dismutase.

**Figure 2 antioxidants-12-00694-f002:**
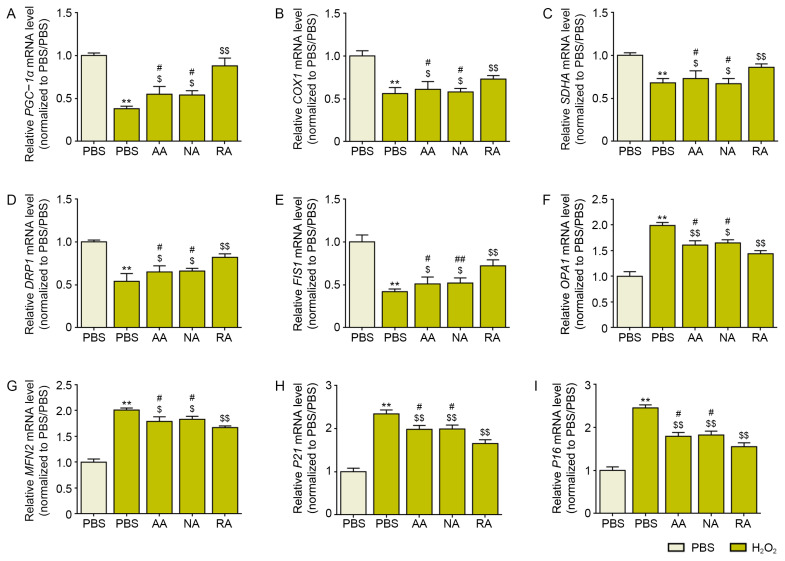
RA treatment decreased mitochondrial dysfunction and cellular senescence in H_2_O_2_-treated keratinocytes. (**A**–**G**) Mitochondrial dysfunction in H_2_O_2_-treated keratinocytes was confirmed by qRT-PCR. The levels of the mitochondrial biogenesis markers PGC-1α (**A**), COX1 (**B**), and SDHA (**C**) were decreased by H_2_O_2_ treatment but increased by RA treatment. The levels of the mitochondrial fission markers DRP1 (**D**) and FIS1 (**E**) were also decreased by H_2_O_2_ treatment but increased by RA treatment. The levels of the mitochondrial fusion markers OPA1 (**F**) and MFN2 (**G**) were increased by H_2_O_2_ treatment but decreased by RA treatment. (**H**,**I**) Cellular senescence in H_2_O_2_-treated keratinocytes was confirmed by qRT-PCR. The levels of the cellular senescence markers P21 (**H**) and P16 (**I**) were increased by H_2_O_2_ treatment but decreased by RA treatment. The data are presented as the mean ± SD of three independent experiments. **, *p* < 0.01, second bar vs. first bar; $, *p* < 0.05 and $$, *p* < 0.01, vs. second bar; #, *p* < 0.05 and ##, *p* < 0.01, vs. fifth bar (Mann–Whitney U test). AA; ascorbic acid, COX; cytochrome c oxidase, DRP1; dynamin-related protein 1, FIS1; mitochondrial fission 1 protein, MFN2; mitofusin 2, NA; niacinamide, OPA1; optic atrophy protein 1, PBS; phosphate-buffered saline, PGC-1α; peroxisome proliferator-activated receptor γ coactivator 1 α, SDHA; succinate dehydrogenase complex and flavoprotein subunit A.

**Figure 3 antioxidants-12-00694-f003:**
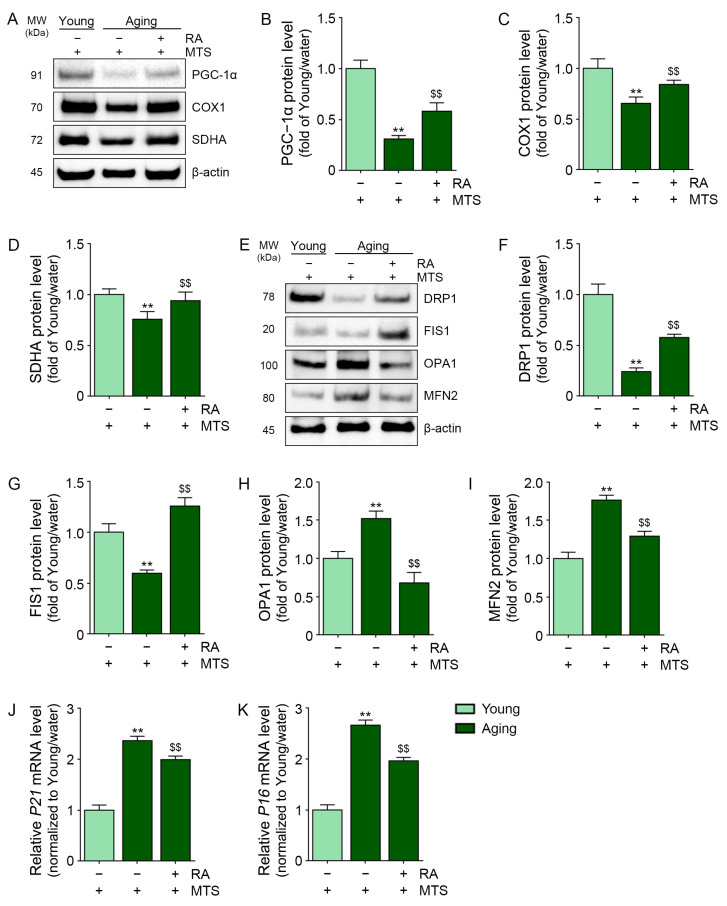
RA treatment decreased mitochondrial dysfunction and cellular senescence in aged mouse skin. (**A**–**I**) Mitochondrial dysfunction in aged mouse skin was confirmed by western blotting. The levels of the mitochondrial biogenesis markers PGC-1α (**B**), COX1 (**C**), and SDHA (**E**) were decreased in aged mouse skin but increased by RA injection. The levels of the mitochondrial fission markers DRP1 (**F**) and FIS1 (**G**) were also decreased in aged mouse skin but increased by RA injection. The levels of the mitochondrial fusion markers OPA1 (**H**) and MFN2 (**I**) were increased in aged mouse skin but decreased by RA injection. (**J**,**K**) Cellular senescence in aged mouse skin was confirmed by qRT-PCR. The levels of the cellular senescence markers P21 (**H**) and P16 (**I**) were increased in aged skin but decreased by RA injection. The data are presented as the mean ± SD of three independent experiments. **, *p* < 0.01, second bar vs. first bar; $$, *p* < 0.01, third bar vs. second bar (Mann–Whitney U test). COX; cytochrome c oxidase, DRP1; dynamin-related protein 1, FIS1; mitochondrial fission 1 protein, MFN2; mitofusin 2, MW; molecular weight, NA; niacinamide, OPA1; optic atrophy protein 1, PBS; phosphate-buffered saline, PGC-1α; peroxisome proliferator-activated receptor γ coactivator 1 α, SDHA; succinate dehydrogenase complex and flavoprotein subunit A.

**Figure 4 antioxidants-12-00694-f004:**
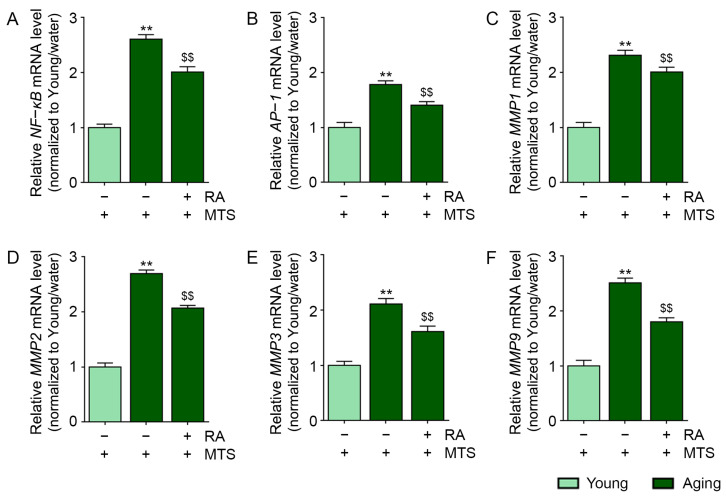
RA treatment decreased NF-κB/AP-1 and MMP1/2/3/9 expression in aged mouse skin. (**A**–**F**) NF-κB/AP-1 and MMP1/2/3/9 mRNA expression in aged skin was confirmed by qRT-PCR. The levels of NF-κB (**A**), AP-1 (**B**), MMP1 (**C**)**,** MMP2 (**D**), MMP3 (**E**)**,** and MMP9 (**F**) were increased in aged skin but decreased by RA injection. The data are presented as the mean ± SD of three independent experiments. **, *p* < 0.01, second bar vs. first bar; $$, *p* < 0.01, third bar vs. second bar (Mann–Whitney U test). AP-1; activator protein 1, MMP; matrix metalloproteinases, NF-κB; nuclear factor kappa-light-chain-enhancer of activated B cells.

**Figure 5 antioxidants-12-00694-f005:**
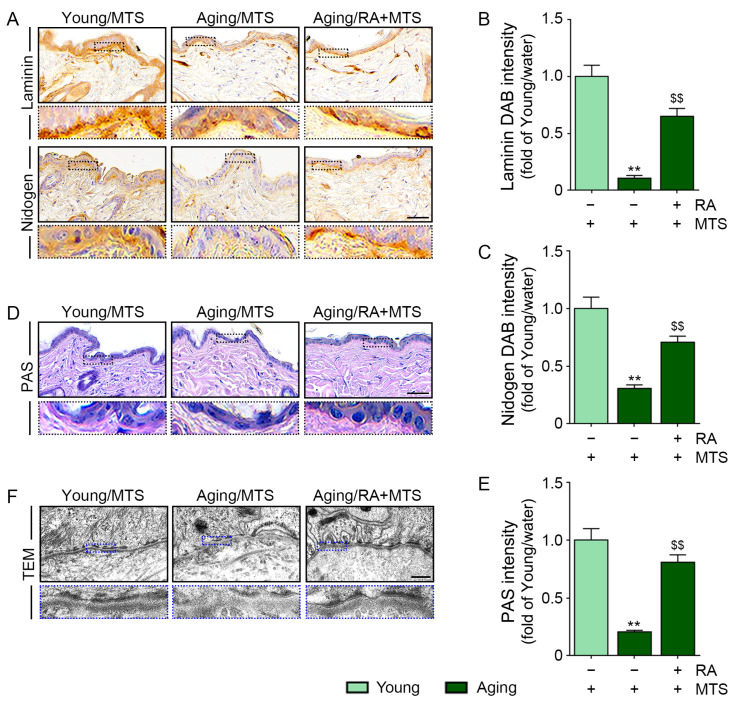
RA treatment restored the BM condition in aged mouse skin. (**A**–**C**) Laminin and nidogen expression was confirmed by immunochemistry (scale bar = 50 µm). The intensities of the laminin (**B**) and nidogen (**C**) signals were decreased in aged mouse skin but increased by RA injection. (**D**,**E**) BM density was confirmed by PAS staining (scale bar = 50 µm). The PAS staining intensity was decreased in aged mouse skin but increased by RA injection. (**F**) The BM condition was confirmed by transmission electron microscopy (scale bar = 5 µm). The hemidesmosomes and lamina densa in TEM images were restored by RA injection. The data are presented as the mean ± SD of three independent experiments. **, *p* < 0.01, second bar vs. first bar; $$, *p* < 0.01, third bar vs. second bar (Mann–Whitney U test). BM; basement membrane, DAB; 3,3′-diaminobenzidine tetrahydrochloride hydrate, PAS; periodic acid-Schiff, TEM; transmission electron microscopy.

**Figure 6 antioxidants-12-00694-f006:**
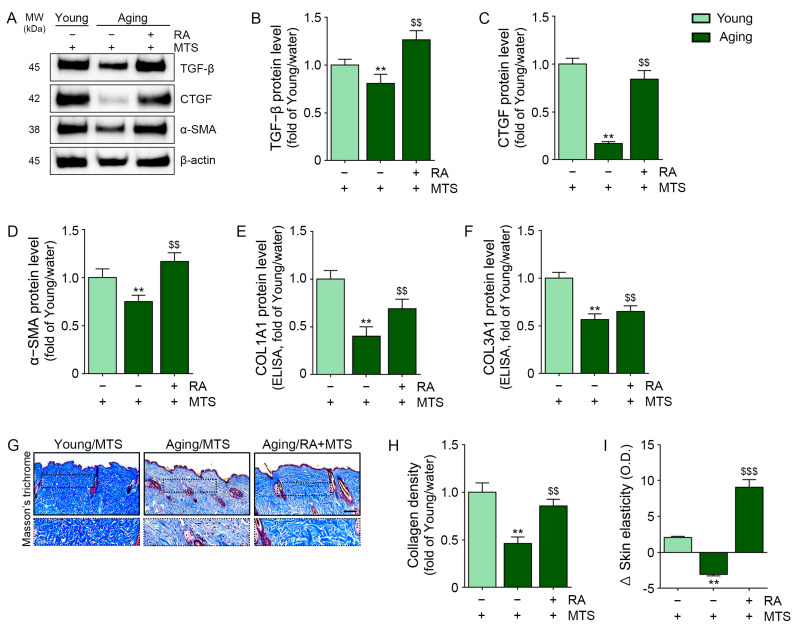
RA treatment increased collagen fiber levels in aged mouse skin. (**A**–**D**) TGF-β, CTGF, and α-SMA expression were confirmed by western blotting. The expression levels of TGF-β (**B**), CTGF (**C**), and α-SMA (**D**) were decreased in aged mouse skin but increased by RA injection. (**E**,**F**) COLI and COLIII expression was confirmed by ELISA. The expression levels of COL1A1 (**E**) and COL3A1 (**F**) were decreased in aged mouse skin but increased by RA injection. (**G**,**H**) Collagen density was confirmed by Masson’s trichrome staining (scale bar = 100 µm). (**I**) Skin elasticity was confirmed. Skin elasticity was decreased in aged mouse skin but increased by RA injection. **, *p* < 0.01, second bar vs. first bar; $$, *p* < 0.01, $$$, *p* < 0.001, third bar vs. second bar (Mann–Whitney U test). α-SMA; α-smooth muscle actin, COL1A1; collagen type I alpha 1, COL3A1; collagen type III alpha 1, CTGF; connective tissue growth factor, ELISA; enzyme-linked immunosorbent assay, MW; molecular weight, TGF-β; transforming growth factor-β.

## Data Availability

All data are contained within the article.

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
