# Peer review of "The Extracellular Matrix Vitalizer RATM Increased Skin Elasticity by Modulating Mitochondrial Function in Aged Animal Skin"

_antioxidants, 2023, doi:10.3390/antiox12030694_

Round 1

Reviewer 1 Report

The manuscript entitled "The extracellular matrix vitalizer RA increased skin elasticity by modulating mitochondrial function in aged animal skin" describes the effects of the proprietary skin regeneration product ECMV-RA booster® (illglobal, Seoul, Korea) on human skin keratinocytes in vitro, as well as, in aged murine skin in vivo. The study is well-designed and the results support the idea that the above product alleviates oxidative stress and reverses a wide range of skin aging markers.

Author Response

Response to Reviewer 1 Comments

The manuscript entitled "The extracellular matrix vitalizer RA increased skin elasticity by modulating mitochondrial function in aged animal skin" describes the effects of the proprietary skin regeneration product ECMV-RA booster® (illglobal, Seoul, Korea) on human skin keratinocytes in vitro, as well as, in aged murine skin in vivo. The study is well-designed, and the results support the idea that the above product alleviates oxidative stress and reverses a wide range of skin aging markers.

Answer. Thank you for review.

Reviewer 2 Report

The Authors presented an article entitled: “The extracellular matrix vitalizer RA increased skin elasticity by modulating mitochondrial function in aged animal skin”.

The manuscript is very interesting, clear, especially when referring to oxidative stress mechanisms, and well organized.

Just few suggestions to the Authors for improving the paper:

In the “Abstract”, please check the English form of the following sentence reported at lines 27-28: “Damage to mitochondrial DNA and mitochondrial fusion markers was increased in aged skin and was decreased by RA”, which should be “Damage to mitochondrial DNA and mitochondrial fusion markers were increased [..]”.

In the “Materials and methods” section, it should be better to report the suppliers of the following reagents: , AA, NA, coenzymes, glutathione, and hyaluronic acid.

In the “Materials and methods” section, subsection “2.3.2. RA treatment”, please, specify the type of water injected when referring to the control.

In figures 5 and 6, please, it should be better to report the scale bars inside the collected images.

Author Response

Response to Reviewer 2 Comments

The Authors presented an article entitled: “The extracellular matrix vitalizer RA increased skin elasticity by modulating mitochondrial function in aged animal skin”. The manuscript is very interesting, clear, especially when referring to oxidative stress mechanisms, and well organized. Just few suggestions to the Authors for improving the paper:

Point 1. In the “Abstract”, please check the English form of the following sentence reported at lines 27-28: “Damage to mitochondrial DNA and mitochondrial fusion markers was increased in aged skin and was decreased by RA”, which should be “Damage to mitochondrial DNA and mitochondrial fusion markers were increased [..]”.

Answer 1. As your recommendation, we revised the sentence of abstract.

Abstract, Page 1, Line 27 in revised manuscript

Damage to mitochondrial DNA and mitochondrial fusion markers were increased in aged skin and decreased by RA. The levels of mitochondrial biogenesis markers and fission markers were decreased in aged skin and increased by RA.

Point 2. In the “Materials and methods” section, it should be better to report the suppliers of the following reagents: , AA, NA, coenzymes, glutathione, and hyaluronic acid.

Answer 2. As your recommendation, we added the material information in supplementary manuscript.

Table S1, Page 2, Line 2 in revised supplementary manuscript

Reagent

Company

Catalog no.

Dilution rate

Ascorbic Acid

Sigma-Aldrich

A92902

0.25%

Niacinamide

Sigma-Aldrich

72340

0.25%

Coenzyme A

Sigma-Aldrich

C4282

0.00002%

Glutathione

Sigma-Aldrich

G6013

0.00005%

Sodium Hyaluronate

Daejung Chemicals & Metals Co. Ltd.

7848-4440

0.20%

Point 3. In the “Materials and methods” section, subsection “2.3.2. RA treatment”, please, specify the type of water injected when referring to the control.

Answer 3. As your recommendation, we revised the materials and methods.

Materials and Methods, Page 3, Line 139 in revised manuscript

The control was injected with distilled water under the same conditions (Figure S1B).

Point 4. In figures 5 and 6, please, it should be better to report the scale bars inside the collected images.

Answer 4. As your recommendation, we added the scale bar in figure legend.

Figure 5, Page 11, Line 395 in revised manuscript

Figure 5. RA treatment restored the BM condition in aged mouse skin. (A–C) Laminin and nidogen expression was confirmed by immunochemistry (scale bar = 50 µm). The intensities of the laminin (B) and nidogen (C) signals were increased in aged mouse skin but decreased by RA injection. (D, E) BM density was confirmed by PAS staining (scale bar = 50 µm). The PAS staining intensity was decreased in aged mouse skin but increased by RA injection. (F) The BM condition was confirmed by transmission electron microscopy (scale bar = 5 µm). The hemidesmosomes and lamina densa in TEM images were restored by RA injection. The data are presented as the mean±SD of three independent experiments. **, p < 0.01, second bar vs. first bar; $$, p < 0.01, third bar vs. second bar (Mann–Whitney U test). BM; basement membrane, DAB; 3,3′-diaminobenzidine tetrahydrochloride hydrate, PAS; periodic acid-Schiff, TEM; transmission electron microscopy.

Figure 6, Page 12, Line 419 in revised manuscript

Figure 6. RA treatment increased collagen fiber levels in aged mouse skin. (A–D) TGF-β, CTGF, and α-SMA expression was confirmed by western blotting. The expression levels of TGF-β (B), CTGF (C) and α-SMA (D) were decreased in aged mouse skin but increased by RA injection. (E, F) COLI and COLIII expression was confirmed by ELISA. The expression levels of COL1A1 (E) and COL3A1 (F) were decreased in aged mouse skin but increased by RA injection. (G, H) Collagen density was confirmed by Masson’s trichrome staining (scale bar = 100 µm). (I) Skin elasticity was confirmed. Skin elasticity was decreased in aged mouse skin but increased by RA injection. **, p < 0.01, second bar vs. first bar; $$, p < 0.01, third bar vs. second bar (Mann–Whitney U test). α-SMA; α-smooth muscle actin, COL1A1; collagen type I alpha 1, COL3A1; collagen type III alpha 1, CTGF; connective tissue growth factor, ELISA; enzyme-linked immunosorbent assay, MW; molecular weight, TGF-β; transforming growth factor-β.

Reviewer 3 Report

This is an interesting paper on important topic, skin aging.

The authors have used C57BL/6 model for the studies. I have several questions as relates to the model.

While only males were used and females were excluded. Males are more aggressive in comparison to females and tend to bite each other. Also hormonal differences between both sexes may affect the outcome of the studies.

Also, are you sure that you made intradermal but not subdermal injections. In mice it is almost impossible to do intradermal injection using standard techniques and needles.

Finally, in C57BL/6 mice melanogenesis is coupled to the anagen phase of hair cycle. Why this is not mentioned. Is hair cycling changing during aging?

Why melanin pigmentation is not discussed taking into consideration its role in skin physiology and pathology (Frontiers in Oncology 2022;12. DOI: 10.3389/fonc.2022.842496).

Melatonin has anti-aging properties and affect mitochondrial functions in skin cells (Cell Mol Life Sci 74(21), 3913-3925, 2017).

Lastly, when mentioning UVR effects the readers would appreciate information on diverse homeostatic actions of the UVR on cutaneous and systemic  homeostasis (Endocrinology 159(5), 1992-2007, 2018).

As relates to the methodology, it is sound and I do not have a major critique here.

Author Response

Response to Reviewer 3 Comments

This is an interesting paper on important topic, skin aging. The authors have used C57BL/6 model for the studies. I have several questions as relates to the model.

Point 1. While only males were used and females were excluded. Males are more aggressive in comparison to females and tend to bite each other. Also hormonal differences between both sexes may affect the outcome of the studies.

Answer 1. We agree with your comments. We observed every day during the animal experiment period, and aggressive animals were separated from the cage or removed from the group. In addition, we are aware that differences in hormones can affect the results, and only males were used because females have severe hormonal changes during the estrus cycle, which can adversely affect the experiment. However, since I agree with your comment, I will try to compare it with female rats in a future study.

Point 2. Also, are you sure that you made intradermal but not subdermal injections. In mice it is almost impossible to do intradermal injection using standard techniques and needles.

Answer 2. We used MTS with a constant depth of 0.5 mm. The depth of the mouse's dermal layer is about 0.7 mm (ref. Wei JCJ, Edwards GA, Martin DJ, Huang H, Crichton ML, Kendall MAF. Allometric scaling of skin thickness, elasticity, viscoelasticity to mass for micro-medical device translation: from mice, rats, rabbits, pigs to humans. Sci Rep. 2017;7(1):15885.), and we used an injection needle with a depth of 0.5 mm, so it was written as intradermal injection. More detailed information on MTS has been added to materials and method.

Materials and Methods, Page 3, Line 135 in revised manuscript

2.3.2. RA treatment

To determine whether aged animal skin was affected by RA treatment, 12-month-old aging mice were injected intradermally with RA (100 μL/cm2/day) twice every two weeks using a microneedle therapy system (MTS; Derma-Q Gold 0.5 mm, DONGBANG medi-care, Seongnam, Korea). The control was injected with distilled water under the same conditions (Figure S1B).

Point 3. Finally, in C57BL/6 mice melanogenesis is coupled to the anagen phase of hair cycle. Why this is not mentioned. Is hair cycling changing during aging? Why melanin pigmentation is not discussed taking into consideration its role in skin physiology and pathology (Frontiers in Oncology 2022;12. DOI: 10.3389/fonc.2022.842496). Melatonin has anti-aging properties and affect mitochondrial functions in skin cells (Cell Mol Life Sci 74(21), 3913-3925, 2017). Lastly, when mentioning UVR effects the readers would appreciate information on diverse homeostatic actions of the UVR on cutaneous and systemic homeostasis (Endocrinology 159(5), 1992-2007, 2018).

Answer 3. As your recommendation, melanogenesis via UV radiation is one of main issue during skin aging. However, we tried to focus on collagen synthesis changes during aging and performed experiment to evaluate whether RA increased collagen synthesis during aging in this manuscript. The experiment to evaluate whether RA could decrease aging related increased melanogenesis should be performed in the future, since the amount of experiment for evaluating melanogenesis are also large.

Point 4. As relates to the methodology, it is sound and I do not have a major critique here.

Answer 4. Thank you for review.

Reviewer 4 Report

The paper is interesting and the results are promising in the inhibition of age-related decrease of skin elasticity and mitochondria impairment, I have one important remark concerning the whole text.

1. Please disclose the actual meaning of  the extracellular matrix vitalizer RA. I cannot find the definition of this abbreviation, and many repetitions of the experiments and deeper study on the described phenomena are not possible without fully dissecting the abbreviation. Other abbreviations are properly defined in the text or in the abbreviations section.

2. Consequently, the composition of the extracellular matrix vitalizer RA may not be fully repeteable in subsequent experiments and the observed effects. The composition should be defined and additionally commented.

3. "The extracellular matrix vitalizer RA (ECMV-RA booster® , illglobal, Seoul, Korea) 96
contains various antioxidants, such as AA, NA, coenzymes, glutathione, and hyaluronic 97
acid" is not precisely defined, and it would be really difficult to repeat these experiments, or design new ones.

4. The aged mice were 12 months old. Is this criterion enough? Is the aging of the mice according to the observations gradual or abrupt? What is the delimitation of the age, below which the skin cannot be determined as aged?

5. Are there other means to applicate RA besides the injection, which may be considered as invasive? E.g. topical?

Author Response

Response to Reviewer 4 Comments

The paper is interesting and the results are promising in the inhibition of age-related decrease of skin elasticity and mitochondria impairment, I have one important remark concerning the whole text.

Point 1. Please disclose the actual meaning of the extracellular matrix vitalizer RA. I cannot find the definition of this abbreviation, and many repetitions of the experiments and deeper study on the described phenomena are not possible without fully dissecting the abbreviation. Other abbreviations are properly defined in the text or in the abbreviations section.

Answer 1. ‘Extracellular matrix vitalizer RA’ is a trademark of the formula of liquid which had been used our experiment. RA is an abbreviation of ‘regeneration anti-aging’. However, we did not think that use full name since it is a part of trademark. Thus, we marked ‘TM’ for clarifying it is a trademark in the text like below;

Introduction, Page 3, Line 97 in revised manuscript

The extracellular matrix vitalizer RATM (RA, illglobal, Seoul, Korea) contains various antioxidants, such as AA, NA, coenzymes, glutathione, and sodium hyaluronate. Thus, we hypothesized that RA injection could decrease oxidative stress and cellular senescence, which eventually resulted in a reduction in NF-κB/AP-1 and mitochondrial dysfunction in the skin. These reductions led to decreased levels of MMPs, which eventually decreased ECM fiber destruction and BM destruction. RA also increased the levels of TGF-β and CTGF, which eventually increased collagen fiber synthesis. We evaluated the RA-mediated increase in collagen fiber accumulation and decreased in BM destruction via decreased oxidative stress in aged animals. The effects of RA were compared with those of AA or NA injected alone into an aged animal skin.

Point 2. Consequently, the composition of the extracellular matrix vitalizer RA may not be fully repeatable in subsequent experiments and the observed effects. The composition should be defined and additionally commented.

Answer 2. We agree with your comments. As your comment, we added the composition of materials.

Materials and Methods, Page 3, Line 108 in revised manuscript

2.1. Preparation of RA

RA was formulated as a liquid before application. First, AA, NA, coenzymes, glutathione, and sodium hyaluronate were dissolved in distilled water with mixing at 3000 rpm using a high-speed mixer (T.K. Homo Disper, Model 2.5, PRIMIX, Hyogo, Japan). Then, the RA solution was filtered through a 0.2 μm filter (S2GPU11RE, Merck, CA, USA) to remove bacteria. The RA liquid contained 0.25% AA and 0.25% NA (Table S1).

Table S1, Page 2, Line 2 in revised supplementary manuscript

Reagent

Company

Catalog no.

Dilution rate

Ascorbic Acid

Sigma-Aldrich

A92902

0.25%

Niacinamide

Sigma-Aldrich

72340

0.25%

Coenzyme A

Sigma-Aldrich

C4282

0.00002%

Glutathione

Sigma-Aldrich

G6013

0.00005%

Sodium Hyaluronate

Daejung Chemicals & Metals Co. Ltd.

7848-4440

0.20%

Point 3. "The extracellular matrix vitalizer RA (ECMV-RA booster® , illglobal, Seoul, Korea) contains various antioxidants, such as AA, NA, coenzymes, glutathione, and hyaluronic acid" is not precisely defined, and it would be really difficult to repeat these experiments, or design new ones.

Answer 3. We agree with your comments. As your comment, we added the composition of RA.

Materials and Methods, Page 3, Line 108 in revised manuscript

2.1. Preparation of RA

RA was formulated as a liquid before application. First, AA, NA, coenzymes, glutathione, and sodium hyaluronate were dissolved in distilled water with mixing at 3000 rpm using a high-speed mixer (T.K. Homo Disper, Model 2.5, PRIMIX, Hyogo, Japan). Then, the RA solution was filtered through a 0.2 μm filter (S2GPU11RE, Merck, CA, USA) to remove bacteria. The RA liquid contained 0.25% AA and 0.25% NA (Table S1).

Table S1, Page 2, Line 2 in revised supplementary manuscript

Reagent

Company

Catalog no.

Dilution rate

Ascorbic Acid

Sigma-Aldrich

A92902

0.25%

Niacinamide

Sigma-Aldrich

72340

0.25%

Coenzyme A

Sigma-Aldrich

C4282

0.00002%

Glutathione

Sigma-Aldrich

G6013

0.00005%

Sodium Hyaluronate

Daejung Chemicals & Metals Co. Ltd.

7848-4440

0.20%

Point 4. The aged mice were 12 months old. Is this criterion enough? Is the aging of the mice according to the observations gradual or abrupt? What is the delimitation of the age, below which the skin cannot be determined as aged?

Answer 4. We wanted to confirm the effect of RA in middle-aged. According to the Jackson lab (https://www.jax.org/research-and-faculty/research-labs/the-harrison-lab/gerontology/life-span-as-a-biomarker), 12-month mice is equivalent to a human in their 40s. Therefore, we used 12-month-old mice for animal experiments.

Point 5. Are there other means to applicate RA besides the injection, which may be considered as invasive? E.g. topical?

Answer 5. You can also use the topical method. However, we tried to investigate the effect of RA treatment on the dermis in this paper.

Round 2

Reviewer 4 Report

I am fully satisfied with the comments and the final version of the manuscript.